# Structural design and optimization of egg carrier for dynamic egg slit detection platforms

**Ronghua Meng**[1,2,3⊚‡], **Yuxiang Tian** [1,2,3*⊚‡], **Siwei Huang**[1,2,3*⊚]

1 The Hubei Key Laboratory of Hydropower Machinery Design & Maintenance, China Three Gorges University, Yichang, Hubei, China, 2 Yichang Key Laboratory of Robot and Intelligent Systems, China Three Gorges University, Yichang, Hubei, China, 3 Intelligent Manufacturing Innovation Technology Center, China Three Gorges University, Yichang, China

⊚ These authors contributed equally to this work.
‡ RM and YT are joint senior authors.
* 3317118967@qq.com (YT); 2965349457@qq.com (SH)

**Data availability statement:** All data files are available from the figshare

## Abstract

This paper presents an improved egg carrier for supporting common types of eggs during an egg slit inspection platform opeations. The carrier conveys the eggs forwards to the inspection position. The egg carrier consists of three basic modules and has the advantages of adaptability, adjustability, low maintenance and manufacturing costs, and stable and efficient operation. First, the egg carrier functions are analysed, and preliminary design concepts and solutions are proposed via TRIZ theory. Second, the egg carrier structure is designed, the material properties are analysed, and key parameters such as deformation are simulated by SolidWorks and Ansys Workbench. Finite element and fatigue analyses are carried out for the key parameters of the structure, and deformation, fatigue and other data that appear in the actual operation process are obtained. On the basis of stable egg carrying, a carrying motion mathematical model is established, and the genetic algorithm (GA) is chosen to optimize the structural parameters. Finally, motion simulation is carried out for the improved egg carrier, which verifies the optimized structure reasonableness.

## 1. Introduction

Egg products are extremely common in daily dietary life, and their food safety is closely related to human health. In general, cracks and contaminants on eggshells can lead to serious food safety problems for poultry and egg products [1]. The effective detection of eggshell cracks can improve product quality. However, at present, the commonly used traditional manual testing method is inefficient and has a high error rate. Some enterprises use automatic testing equipment with low efficiency, poor stability and high cost to meet market demand. Therefore, there is an urgent need to develop fully automated and highly efficient automatic eggshell crack testing equipment to address market demand while reducing costs.

When eggs enter and leave a factory prior to entering the sales market, preprocessing and screening work need to be completed. These dynamic methods include egg crack detection,

database (DOI: https://doi.org/10.6084/m9.figshare.28468568.v1)

**Funding:** This work is financially supported by the National Natural Science Foundation of China (Grant No. 51975324, Grant No. 52075292) and the Natural Science Foundation of Hubei Province (Grant No. 2022CFC033). The funders had no role in study design, data collection and analysis, decision to publish, or preparation of the manuscript.

**Competing interests:** The authors have declared that no competing interests exist.

egg cleaning, and detection of whether the eggs are fertilized, all of which require stable and efficient egg carriers. Because eggshells have low strength and break easily, improving the material and structure of egg trays is necessary to ensure high efficiency and stability during transportation. Moreover, structural analysis is needed to ensure that the egg carrier rod can bear the torque generated during the transportation process. The egg carrier should prevent problems such as sticking, dropping or leaking eggs, causing the carrier to damage the egg-shells. Therefore, it is important to optimize the structure of this part to adapt it to increasing production volume. Thus, this paper proposes an improved egg carrier.

This paper is organized as follows. Section 1 describes the problem through TRIZ theory. Section 2 proposes a solution for the optimisation of key components through genetic algorithms. In Section 3, computational experiments are carried out on the improved egg bearer through ANSYS Workbench and SolidWorks. The comparative experiments between the old egg carrier and the improved egg carrier are carried out by kinematic analysis, which proves that this research solves the main problems that exist at present. Section 4 describes management insights, including the applicability of the model, the simplicity of the practice, and how the thesis benefits production. Finally, the conclusion presents the results of the study, the limitations of the study, and recommendations for future research.

## 2. Literature review

### 2.1. Related studies

The field of mechanical design optimization has undergone significant advancements in recent years, with numerous methods emerging to address complex engineering challenges. These methods range from geometric optimization to probabilistic design optimization, each offering unique insights and solutions. For example, Bai Yunfei et al. [2] proposed a gap geometry optimization method, which significantly improved the performance of deep-sea electric manipulators by optimizing the compensating oil viscous power. This approach highlights the potential of geometric modifications in enhancing mechanical efficiency and reliability.

In another study, Feng Mei et al. [3] developed a new inverse kinematics solution based on repetition error compensation, which has been instrumental in improving the accuracy and efficiency of robotic systems. This method addresses the inherent errors in robotic movements, providing a robust framework for high-precision applications. Hashem Mehanpour et al. [4] further expanded the optimization landscape by introducing probabilistic design optimization (PDO) and reliability-based probabilistic design optimization (RBPDO). Yang Chengxing et al [5] used a two-step optimization method to improve material utilization. These methods have been particularly useful in assessing and enhancing the reliability of truss structures, ensuring their performance under uncertain conditions.

Topology optimization has also gained prominence in structural design, with Selvaraj Ganeshkumar et al. [6] proposing a method for designing and manufacturing air compressor pistons using topology optimization and metal additive manufacturing technology. This approach not only optimizes structural performance but also leverages advanced manufacturing techniques to achieve complex designs. Yang Mei et al. [7] presented an innovative systematic approach for designing large-scale complex mechanical systems, emphasizing the importance of multidisciplinary methods in addressing intricate design challenges.

Simulation-based design has become increasingly important, especially in the context of structural optimization. Gao Changqing et al. [8] performed kinematic analysis of a robot gripper, using SolidWorks to build the model and further optimize the structure. This study demonstrates the effectiveness of simulation tools in validating and refining designs. Ji Leifan et al. [9] combined SolidWorks and ANSYS Workbench to perform structural analysis and

optimization of a study table lifting device, highlighting the potential of integrated simulation environments in enhancing design accuracy and efficiency.

Dong Yanhao et al. [10] established a parametric model of a tension machine frame using SolidWorks and optimized the frame design with ANSYS Workbench. Their work highlights the practical application of simulation-based optimization in industrial settings. Qiang Hongbin et al. [11] simulated the movement of a ship, whereas Ning Tingzhou et al. [12] focused on the movement of the horizontal sealing device of a bulk packaging machine using SolidWorks. These studies underscore the versatility of simulation tools in analysing and optimizing dynamic systems.

Qing Yuye [13] conducted a kinematic simulation analysis of rotary body parts, providing valuable insights into the behaviour of rotating components. Cervantes Culebro Hector et al. [14] addressed the challenges of high-speed robot pick-and-place tasks by optimizing the vibration effect, whereas Niu Guojun [15] proposed a modular design to reduce component coupling. Avikal S et al. [16] utilized fatigue life analysis [17,18] to assess the performance of front axle beams, emphasizing the importance of durability in mechanical components.

The integration of optimization algorithms with simulation tools has further enhanced the capabilities of design optimization. Lu Tianhai et al. [19] proposed a neighbourhood mixed mean-centred inverse learning particle swarm algorithm (NHCOPSO) based on optimizing the PID controller, demonstrating the potential of advanced algorithms in improving control systems. Yang Zedong [20] used a genetic algorithm (GA) for elite design, whereas Chen Yulian et al. [21] applied a GA to solve optimization models quickly. These studies highlight the effectiveness of the GA in finding optimal solutions for complex design problems.

## 2.2. Research gap analysis and contributions

Despite extensive research in the field of mechanical design optimization, several gaps remain, particularly in the context of egg carrier design for dynamic egg slit detection platforms. While previous studies have explored various optimization techniques [22], few have focused on the specific challenges associated with egg transportation. For example, traditional manual testing methods for eggshell cracks are inefficient and prone to errors, and the existing automatic testing equipment often suffers from low efficiency, poor stability, and high costs [1]. This highlights the need for fully automated and highly efficient eggshell crack testing equipment that can meet market demands while reducing costs.

The current egg carriers used in industry often face issues such as egg drop, cracked eggs during transportation, and unstable egg rolling. These problems not only affect the efficiency of the production process but also compromise the quality of the final product. The existing designs lack the necessary structural optimization to address these challenges [23], and there is a need for innovative solutions that can enhance the stability and efficiency of egg transportation.

This study aims to fill these gaps by proposing an improved egg carrier design that leverages advanced optimization techniques. By addressing these gaps, this research provides a robust framework for the design and optimization of egg carriers, contributing to advancements in the field of mechanical engineering.

## 3. Problem description

### 3.1. Analysing problems via TRIZ theory

To better analyse the problems that may occur when the improved egg carrier meets the above functional requirements, TRIZ theory [24], also known as inventive problem solving theory, is applied to determine the product design position. Research has focused on the use of existing

egg carriers. An egg carrier will occasionally drop eggs during the carrying process. If eggshells are broken during egg transport, they are not able to roll forward stably, and other problems can occur. Therefore, on the basis of TRIZ theory, this paper obtains the following conflict resolution matrix. The data are shown in Table 1.

The functional requirements and conflict resolution matrix analysis described in Table 1 provide directions for further structural design optimization.

## 3.2.  Structural design of the egg carrier rod

As a key part connecting the egg tray and friction wheel in the egg carrier, the egg carrier rod acts as a bridge and link. The main function of the egg carrier bar is to carry and transfer torque. This ensures that the egg tray and the friction wheel can work smoothly and accurately, thus ensuring the efficient operation of the whole egg carrier device.

The egg carrier rolls forward through the friction generated by the contact between the friction wheel and the friction guide. In this process, the egg carrier rod needs to have sufficient strength and structural stability to ensure that there will be no breakage or deformation during operation. Moreover, it is also necessary for the egg carrier to transmit torque effectively so that the egg tray can roll forward stably while carrying the eggs.

The egg carrier rod also needs to be stabilized during torque transfer. If a simple interference fit is used between the friction wheel and the egg carrier rod, defects such as inaccurate rolling and relative displacement during transportation may occur. To solve these defects, this design improves the matching method between the egg carrier rod and the friction wheel. The pin axial fixing method is adopted to ensure that the friction wheel and the egg carrier rod do not experience relative displacement, which ensures the transmission of torque and the egg carrier stability during operation.

## 3.3.  Structural design of the friction wheel

The friction wheel, as a key component that converts friction force into torque, has an important role in the structure of the egg carrier. The friction wheel in this design consists of three main components: the wheel body, friction ring and pin. The wheel body is made of carbon steel to ensure its stability over a long period. The friction ring covering the surface of the wheel body is connected via a plug-and-play method to provide a sufficient friction coefficient for effective friction conversion.

During egg carrier assembly, the friction wheel is mounted on the egg carrier rod, which is fixed by means of a pin to ensure the stability of the connection and the effective transmission of torque. This connection is not only simple and reliable but also easy to maintain and replace.

Overall, the friction wheel plays a key role in the egg carrier. Through the design and optimization of its structure, the friction force can be effectively converted into torque, thus driving the rolling of the egg carrier in the process of operation.

Table 1.  Conflict resolution matrix based on TRIZ theory.

| Description of the problem | Improvement parameters | Deterioration parameters | Possible Inventive Principles |
|---|---|---|---|
| Egg drop during transportation | Egg Carrier Stability | Structural complexity | Searching for the optimal parameters of a structure using an intelligent optimization algorithm |
| Cracked eggs during transportation | The material is too hard | Costs | Use of flexible materials |
| Eggs do not roll steadily forwards during transportation | Structural Rationality of Egg Carrier | Weights | Optimization of structural parameters using simulation software |

### 3.4. Structural design of the egg tray

As a key component of egg carriers, egg trays have a delicate design and a wide range of applications. The egg tray in this design is made of food-grade epoxy resin, which ensures safety and hygiene in the production process. Second, the soft material enables the egg tray to provide a good cushioning effect when carrying eggs, reducing the shock and vibration received by the eggs during efficient transportation or storage. Finally, the properties of the epoxy resin material allow the egg tray to maintain stable performance at different room temperatures, ensuring egg safety and meeting the needs of egg carriers operating in different working environments. In addition, there must be enough friction between the egg carrier and the bar to transmit torque to make the eggs roll smoothly and steadily.

Overall, as a food-grade component for carrying eggs, the egg tray is designed with stability, convenience, safety and other factors in mind. This design involves optimizing the structure to improve the stability and load-bearing capacity of the component, as well as the ability to adapt to different types of eggs.

To summarize, the improved egg carrier designed in this study considers stability, practicality, convenience, safety and innovation. By combining GA optimization and motion simulation analysis, the efficiency and stability of transportation can be improved to meet the demands of factory production and transportation.

## 4. Solution approach

### 4.1. Development of a mathematical model for egg carrier optimization

The egg tray material is food grade epoxy resin, so the main production and processing of this component for the injection moulding process can better fill the mould to avoid air bubbles. During the demoulding process, to remove the egg tray smoothly and completely, the design of the egg tray structure is simplified to a cylindrical structure consisting of a straight line and a section of the closed curve rotation arc.

In the GA optimization design process, the egg is simplified to an ellipse. A review of the data revealed that the average length of the long half of the egg was 32 mm, and the average length of the short half was 23.5 mm.

The equation for the ellipse is:

$$\frac{x^2}{32^2} + \frac{y^2}{23.5^2} = 1$$

$x$ is the long half-axis of the ellipse.

$y$ is the short half-axis of the ellipse.

In the mathematical model design process, the end point of the egg clamping arc of the egg conveyor is designed as point b. To meet the design requirements, the length of the egg conveyor should be as small as possible. At the same time, it must stably plate eggs. The diameter of eggs commonly on the market is 45–55 mm. Analysing the spacing of the old egg carriers, which are currently used in bulk in factories, the following data was obtained. To support the eggs, the spacing between the egg carriers should be less than 65 mm. Therefore, the maximum diameter of egg holders should be less than 32.5 mm.

In this study, the main optimization is the structure of the egg carrier; the shape and performance of the egg holder are optimized, and the values for the length and diameter of the old egg carrier are reasonable. Therefore, the diameter and length of the old egg carrier are used, and the coordinates of point b are set to (40, 27.5).

Afterwards, the mathematical model of the circular arc is obtained as:

$$(40 - x_c)^2 + (27.5 - y_c)^2 = r^2$$

where ($x_c$, $y_c$) are the coordinates of the centre of the circle and where $r$ is the radius of the arc specified in the experimental algorithm.

The volume of the column enclosed by an arc rotated about the x-axis for one cycle is:

$$V = \pi \int_{hw}^{40} \left( r^2 - (40 - x_c)^2 \right) dx$$

Therefore, the volume equation for this egg carrier is as follows:

$$TV = 2V + 2\Pi hw^2 (40 - hw)$$

where $hw$ is the distance from the bottom end of the ellipse to the x-axis, i.e., the thickness of the platform at the bottom of the egg carrier. We selected more than 6 mm in the design process to ensure that the egg carrier egg tray met the design requirements in terms of strength.

To achieve egg carrier stability during the transportation of eggs and ensure that the eggs are subjected to low contact forces, problems such as cracked or jammed eggs are avoided. For our structural optimization, we redesigned the egg tray, including the middle platform and the two sides of the arc-shaped support structure. To realize this function, it is necessary to ensure that while the middle platform supports the eggs, both sides of the arc have contact with the eggs to stabilize and provide part of the friction.

## 4.2. Finding the position of the centre of a circle via an experimental algorithm

The experimental algorithm is designed such that there is a tangent point to the ellipse without determining the centre of the circular arc. The algorithm uses the bisection method of computation to determine the location of the centre of the arc of the circle where the tangent point exists and yields the experimental results shown in Fig 1.

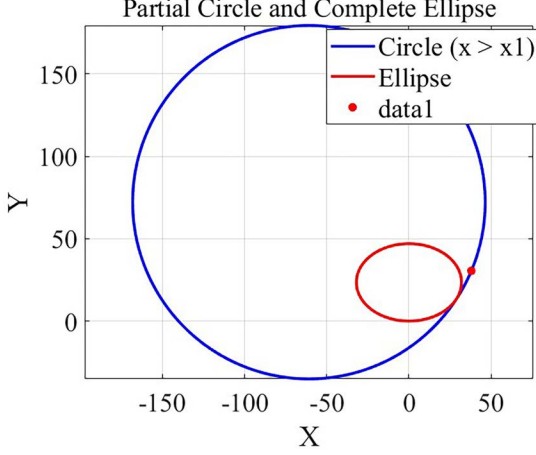

**Fig 1. Schematic diagram of the experimental algorithm.**

This result shows that the location of the centre of the arc of a circle with a tangent point to the ellipse and passing through point b can be found accurately via the bisection method for a certain radius of the circle.

### 4.3. Designing genetic algorithm variables

The selection of the optimization design variables directly affects the final result of the optimization objective function, which should be determined according to the structural parameters of the egg carrier. The following five parameters are selected as the variables for the optimization design: the horizontal coordinates of the arc origin x, the vertical coordinates of the arc origin y, the radius of the arc r, the height of the egg carrier $hw$, and the height of the arc egg carrier position.

$$X = [x1, x2, x3, x4, x5] = [x, y, r, hw, highmin]$$

Range determination for these variables is performed.

(1) Arc radius

According to the egg carrier structure design, the radius is constrained as follows:

$$50 < r < 200$$

(2) Coordinates of the arc origin

The coordinates of the origin of the arc cannot be determined via mathematical modelling. Using MATLAB with the bifurcation method of the arc over point b and a radius of r for the solution, the centre of the circle coordinates can be obtained to meet the requirements.

(3) Support the height of the eggs

The size of the aperture can be determined according to the egg tray selection of materials for food-grade epoxy resin and egg-bearing rods. The minimum diameter of the egg tray is 6 mm, and the egg height constraints are as follows:

$$6 < hw < 27.5$$

(4) Arc-bearing poultry egg height

Owing to the need to ensure the stability of the egg in the design process, the arc structure of the egg tray must play a role in supporting the egg. To support stability, the constraints are as follows:

$$highmin > 5$$

### 4.4. Analysis of results

Previous experimental procedures show that it is feasible to use the GA to try different sizes of arc radii for optimization. On the basis of the optimization mathematical model and the introduction of the optimization algorithm, the algorithm is written.

The GA uses a roulette wheel selection strategy, which is based on the proportion of each individual fitness value to the sum of all individual fitness values. First, it is necessary to

calculate the sum of all individual fitness values, i.e., 'totalFitness'. Afterwards, the probability of selection of each individual is calculated as a probability, i.e., the individual fitness divided by totalFitness. Then, the cumulative probability 'cumProbs' is calculated, i.e., the cumulative sum of the probability of selection of an individual. Finally, for each new position in the population, a random number 'randNum' is generated. The first position in cumProbs that is greater than or equal to randNum is found, and the individual corresponding to that position is selected.

In the crossover process, the GA uses a linear crossover method to generate new candidate solutions. The crossover strategy is used to generate new offspring individuals by selecting some gene fragments from two parent individuals and then combining these fragments in a certain way. The code is a random linear crossover between neighbouring individuals. First, two neighbouring individuals are randomly selected as parents (parent1 and parent2). Second, whether the crossover is performed according to the crossover probability (crossRate) is determined. If the judgement concludes that crossover is needed, the crossover function is subsequently called to generate two children (child1 and child2). Finally, the offspring replace the parent individuals in the original population through a generational replacement strategy

The code determines, for each individual in the population during the mutation process, whether to mutate on the basis of the mutation probability (mutateRate). If mutation is needed, the mutation function is called to mutate individuals randomly. The mutated individuals replace the individuals in the original population.

Afterwards, the fitness code is designed such that the GA randomly generates the value of the radius. Therefore, the size of the radius is known when the fitness is calculated. The location of the centre of the circle needs to be determined such that the rotational volume of the upper semicircle is enclosed by this centre. A given radius r and a straight line segment are minimized while satisfying the condition that the height of the intersection point exceeds a given minimum height. By minimizing the volume of rotation (i.e., maximizing the fitness value), the optimal location of the centre of the circle (-13.943655, 61.359327) can be found.

After running the simulation, the locally optimal solution is obtained, and the values of the calculated design variables are shown in Fig 2.

As shown in Fig 3, during the iteration process, the optimization results begin to converge, and the fitness has improved considerably, indicating the superiority of the GA in solving the problem.

In combination with the GA, this improved egg carrier can provide not only bottom support but also side support for eggs. At the same time, it is also optimized in terms of volume compared with the existing egg carrier. The volume of the existing egg carrier is approximately 0.055 $m^3$, whereas the volume of the improved egg carrier is approximately 0.045 $m^3$, which is 18.18% less.

## 5. Computational experiment and case study

### 5.1. Validation of egg carrier modelling and related structural optimization

After optimization by the GA, the structure of the egg holder is adjusted, and the key parts are verified via simulation. The improved egg carrier is modelled in 3D with SolidWorks. The SolidWorks model is adjusted on the basis of MATLAB-calculated data to make the model more practical and aesthetically pleasing. The static strength of the key part of the egg carrier is subsequently checked via the finite element analysis software ANSYS. Before the finite element analysis, the parts are simplified under the premise of quality assurance, imported into ANSYS software, and assigned material properties and constraints. The egg-bearing

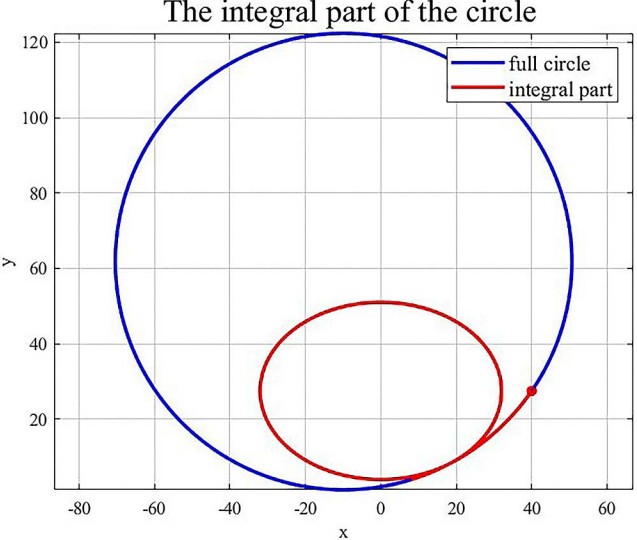

**Fig 2. Schematic diagram of the local optimal solution.**

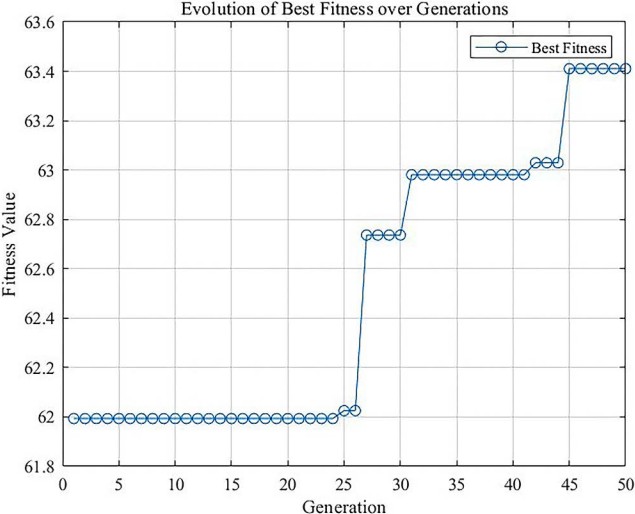

**Fig 3. Optimization results for fitness.**

rod is made of EN AW-2007 aluminium alloy, which has high strength and toughness, good mechanical properties, and corrosion resistance and can maintain stable performance in a variety of environments, as shown in Table 2.

The material for the friction ring is nylon 101, which has good mechanical properties, good abrasion and impact resistance, and high strength and stiffness. The material is also corrosion resistant and has good thermal stability, as shown in Table 3.

The material of the pin and friction wheel body is ordinary carbon steel, and the material is relatively inexpensive. This material has relatively high strength and hardness, good toughness, and a certain range of resistance to deformation and fracture, as shown in Table 4.

**Table 2. Mechanical properties of EN AW-2007 aluminium alloy materials.**

| Densities $g / cm^3$ | Modulus of elasticity $N / m^2$ | Poisson's ratio v | Tensile strength $\sigma b / mpa$ | Yield strength $\sigma$ s/MPa |
|---|---|---|---|---|
| 2.7 | 7e+10 | 0.39 | 370 | 250 |

**Table 3. Mechanical properties of the nylon 101 material.**

| Densities $g / cm^3$ | Modulus of elasticity $N / m^2$ | Poisson's ratio $v$ | Tensile strength $\sigma b / mpa$ | Yield strength $\sigma s / mpa$ |
|---|---|---|---|---|
| 1.15 | 1e+09 | 0.3 | 792 | 600 |

**Table 4. Mechanical properties of the plain carbon steel materials.**

| Densities $g / cm^3$ | Modulus of elasticity $N / m^2$ | Poisson's ratio $v$ | Tensile strength $\sigma b / mpa$ | Yield strength $\sigma s / mpa$ |
|---|---|---|---|---|
| 7.8 | 2.1e+10 | 0.28 | 400 | 220 |

The material chosen for egg trays is food-grade epoxy resin, which has a wide range of applications in food processing and packaging as a special engineering material. The material does not release harmful substances when it is in contact with food, complies with food safety standards, has strong strength and hardness, and has good abrasion and impact resistance. It is formed mainly by injection moulding, which can meet the needs of different shapes, and the processing is convenient and efficient, as shown in Table 5.

**Table 5. Mechanical properties of the epoxy resin materials.**

| Densities $g / cm^3$ | Modulus of elasticity $N / m^2$ | Poisson's ratio $v$ | Tensile strength $\sigma b / mpa$ | Yield strength $\sigma s / mpa$ |
|---|---|---|---|---|
| 1.1 | 2.4e+9 | 0.35 | 280 | none |

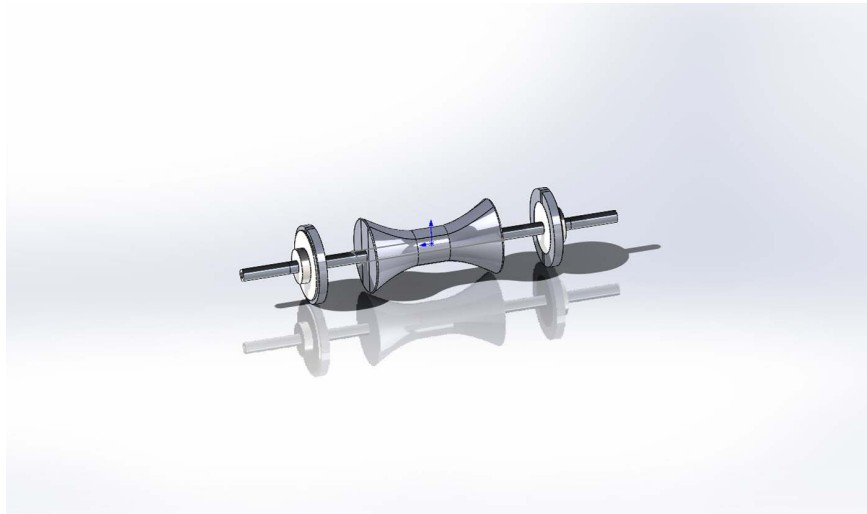

**Fig 4. Improved egg carrier model.**

The components of the egg carrier are simulated via SolidWorks and assembled according to the constraints. The improved egg carrier was obtained as shown in Fig 4.

To illustrate the assembly between the components more intuitively, the model exploded view is shown in Fig 5.

After the model is built, finite element analysis is carried out via SolidWorks simulation software. First, the model is meshed because the structure of the model is not very complex, so the best quality mesh automatically generated by the software can be used. A total of 67179 meshes are generated, as shown in Fig 6.

The definition considers the fact that the egg carrier is subjected to pressure from the transported eggs and the interaction force between the friction wheel and the friction wheel guide under actual working conditions. Therefore, the constraints are constructed such that the pressure on the egg carrier is 2 N. The reaction force on the friction wheel is the weight of the

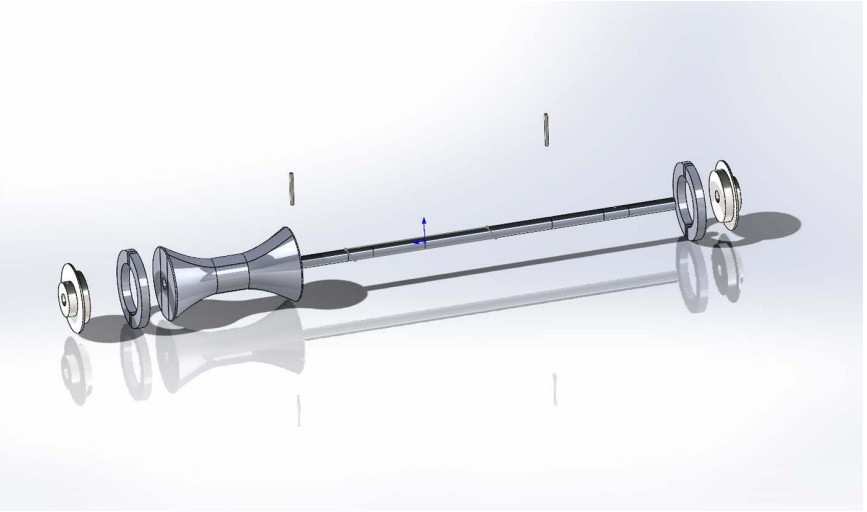

**Fig 5. Exploded view of the improved egg carrier.**

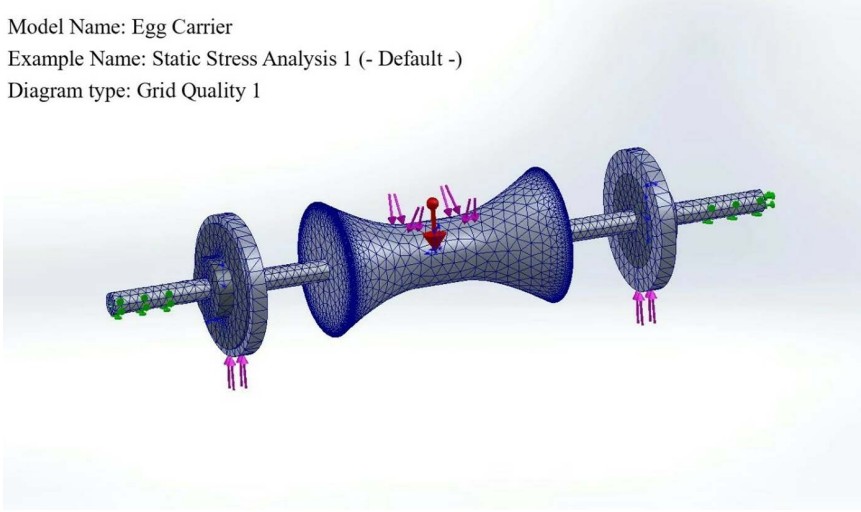

**Fig 6. Automatic generation of mesh diagrams.**

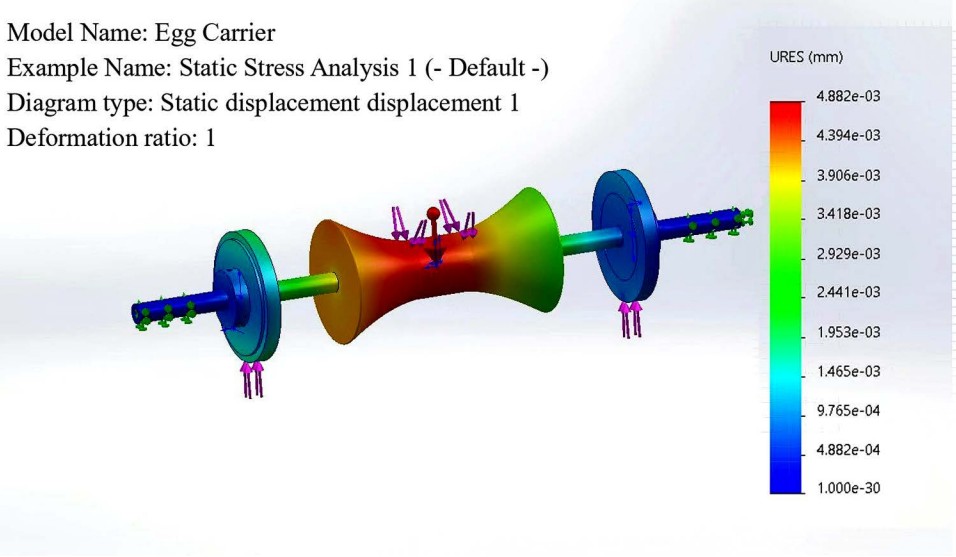

**Fig 7. Net stress analysis static displacement.**

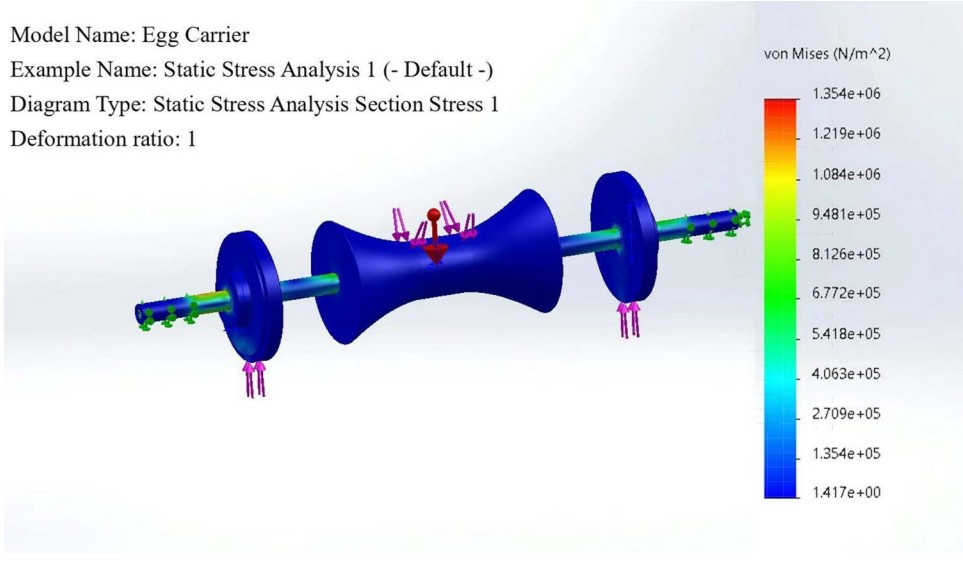

**Fig 8. Static stress analysis section stress.**

egg plus the weight of the egg carrier, which is 6 N. The resultant deformations and stresses on the assembly are calculated, as shown in Figs 7 and 8. The result is that the structure of the assembly meets the needs of the process and that the structure does not produce stress concentrations or large deformations.

The main force object of the structure is the egg-bearing rod. Therefore, ANSYS Workbench software is used to carry out finite element analysis of the egg-bearing rod alone. First, the model is imported and checked via the design method. After checking the model, the constraints are applied to the model. The component is subjected to bending and tensile loads and torque through the friction wheel. The calculation results are shown in Figs 9 and 10.

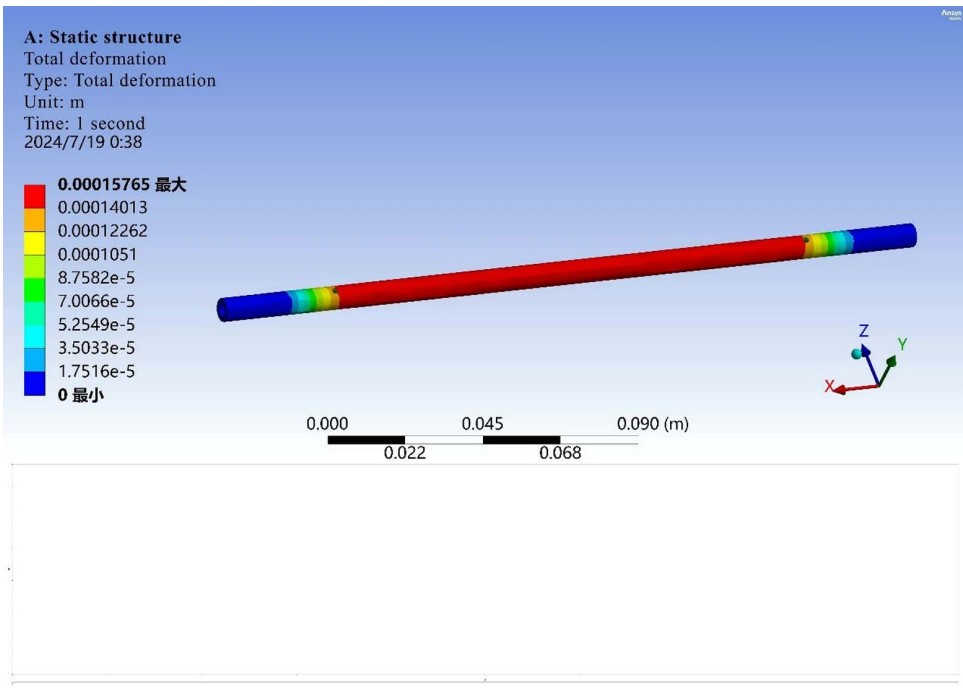

**Fig 9. Total deformation.**

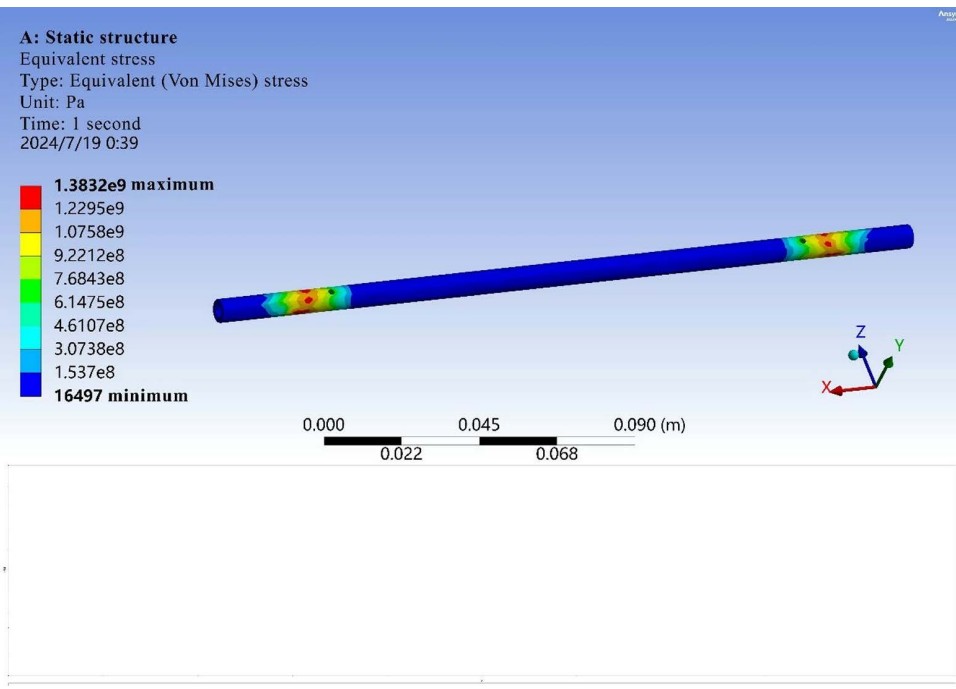

**Fig 10. Equivalent forces.**

Finite element analysis revealed that the part does not experience torsional deformation, fracture or other defects under normal working conditions.

This part of the structure is the main load-bearing structure, and the improved egg carrier is a component for the transportation of eggs. Therefore, operating for long periods in a factory environment is necessary. Under these conditions, a fatigue analysis is required to further determine whether the part can meet the operational requirements.

The fatigue analysis module is imported on the basis of finite element analysis. Since there is no s–n curve for this material in ANSYS software, manually adding the curve to the engineering data is needed. Fig 11 shows the s–n curve of this material.

The results when the fatigue parameter is set for the loading type are exactly opposite. The theoretical curves of constant-amplitude loading and average stress rest are shown in Figs 12 and 13.

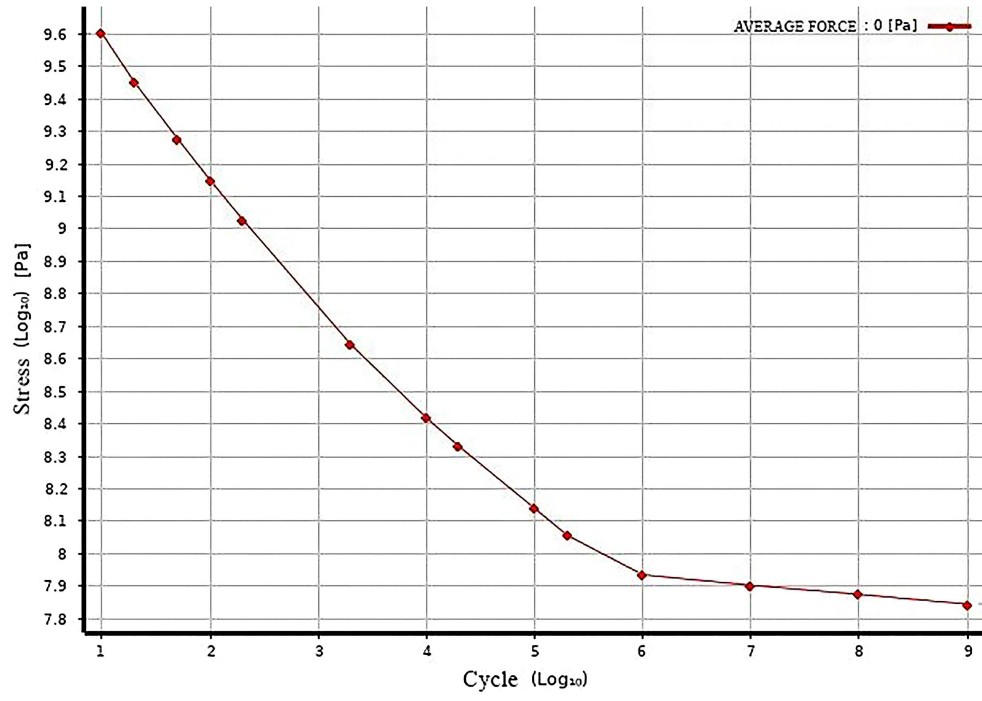

**Fig 11. s–n curves for EN AW-2007 aluminium alloy.**

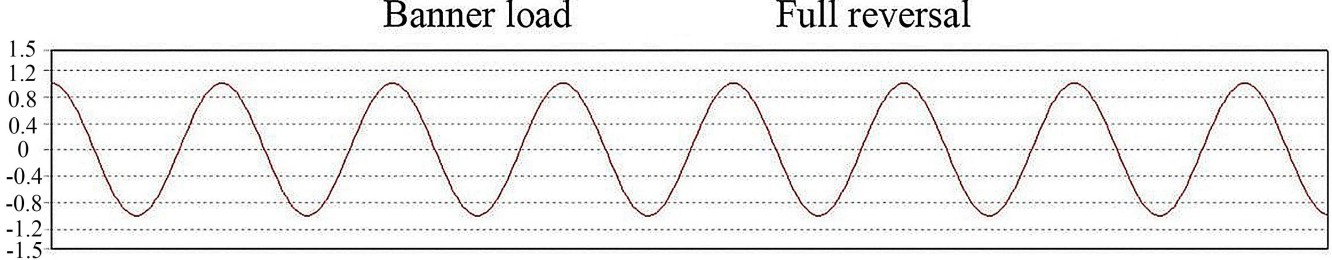

**Fig 12. Banner load chart.**

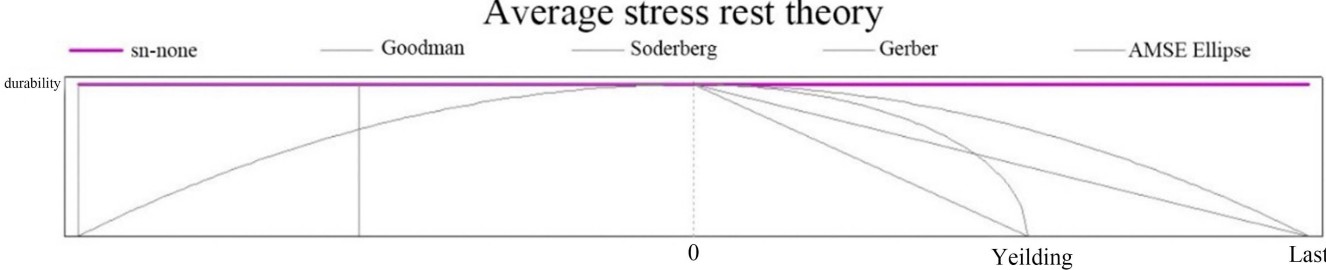

**Fig 13. Theoretical plot of the average resting stress.**

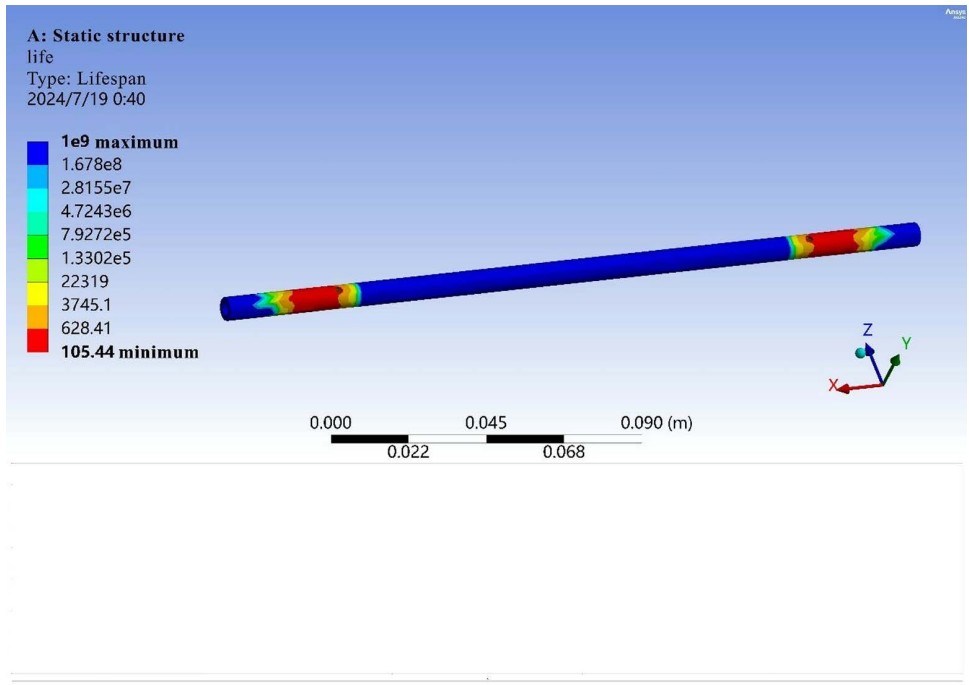

**Fig 14. Schematic diagram of egg-bearing rod life.**

After the relevant parameters are set to simulate a stable transport environment as much as possible, the results are calculated, as shown in Fig 14.

The analysis results indicate that the life of the component can reach 105 cycles, which meets the design requirements.

## 5.2. Egg-bearing kinematic analyses

First, the built 3D model is imported into SolidWorks motion software. The simulation of the egg carrying production line is realized by a linear arrangement, and the position parameters and size specifications of the model are set accurately. The interrelationships between the components are constrained. The eggs designed are subsequently placed on the transportation platform. The adaptability of this improved egg carrier to different kinds of eggs is examined through the fitness of the model, and a constraint relationship is established.

This structural optimization involves optimizing stability and efficiency when eggs are produced. Therefore, without affecting the results, the whole system is operated by adding motors at both ends of the egg carrier. The simulation is shown in Fig 15.

This can be seen intuitively by calculating the angular speed of the improved egg carrier and the displacement of the centre of mass of the egg when the egg carrier is in operation, as shown in Fig 16. The linear displacement of the egg begins to result in defects in unstable movement only when the speed of the improved egg carrier reaches approximately 3,500 deg/s. The old egg carrier has an angular speed of up to approximately 1,200 deg/s.

Therefore, the improved egg carrier can support higher speeds and ensure that the eggs roll steadily forward. Thus, the improved egg carrier is better than the old egg carrier in terms of efficiency.

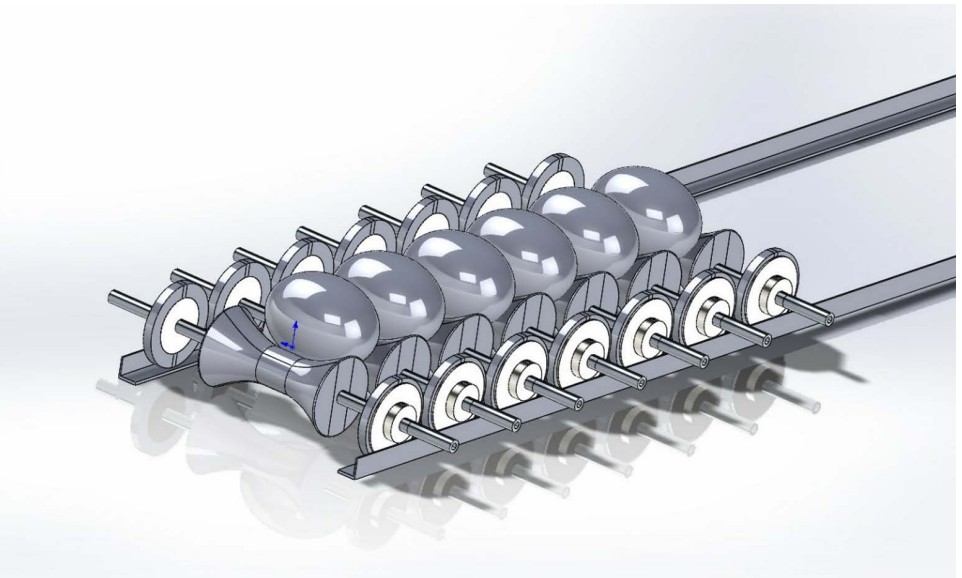

**Fig 15. SolidWorks motion.**

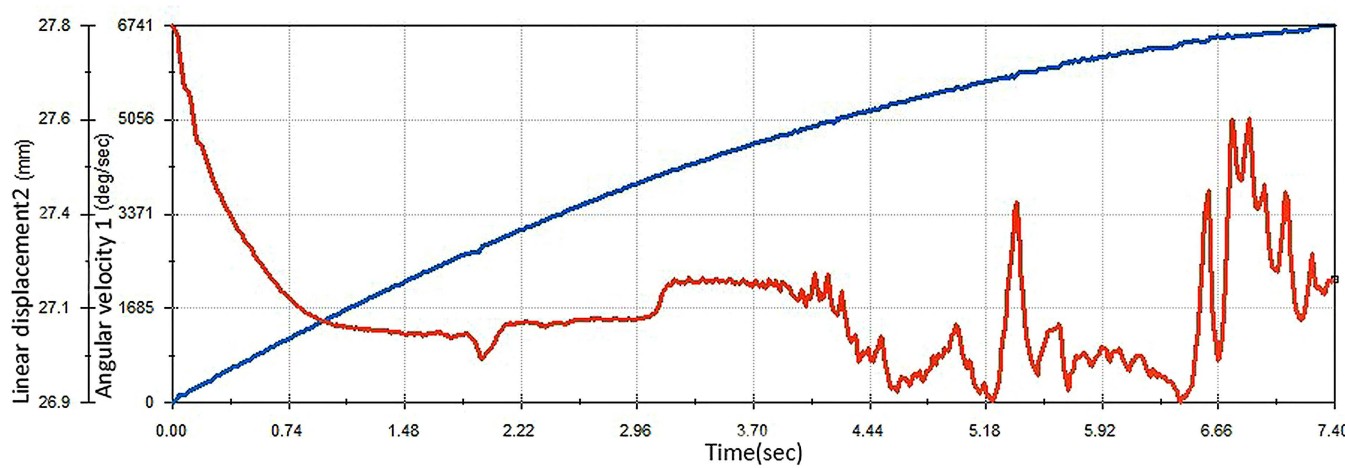

**Fig 16. Relationship between the angular velocity of the improved egg carrier and the linear displacement of the egg.**

To prevent shell cracking, excessive stress on eggs during transportation that causes damage should be minimized. SolidWorks motion and simulation are used to simulate the deformation of the egg during transportation and to calculate the safety factor of the egg. Figs 17 and 18 show the results.

The deformation of the egg during transportation within a reasonable range prevents damage to or cracking of the shell. The minimum safety factor is 3.3.

The kinematic analysis suggests that the improved egg carrier is superior to the old egg carrier in terms of transport speed, i.e., transport efficiency; at the same time, the safety factor of the egg during transport is within the safe range and does not cause defects or ruptures in the eggs.

## 5.3. Experiment and result analysis

Both the old egg carrier and the improved egg carrier were simulated in this designed comparison experiment. First, the contact forces on the improved egg carrier and the old egg carrier during transport were analysed, as shown in Figs 19 and 20.

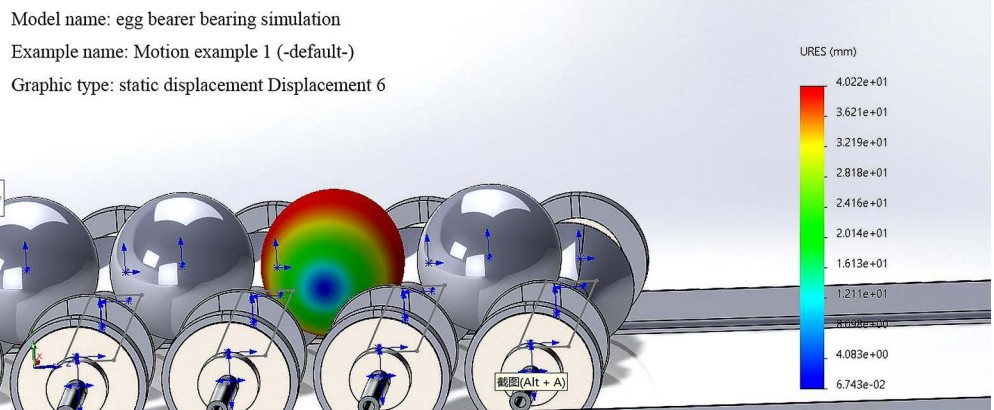

**Fig 17. Deformation of eggs during transportation.**

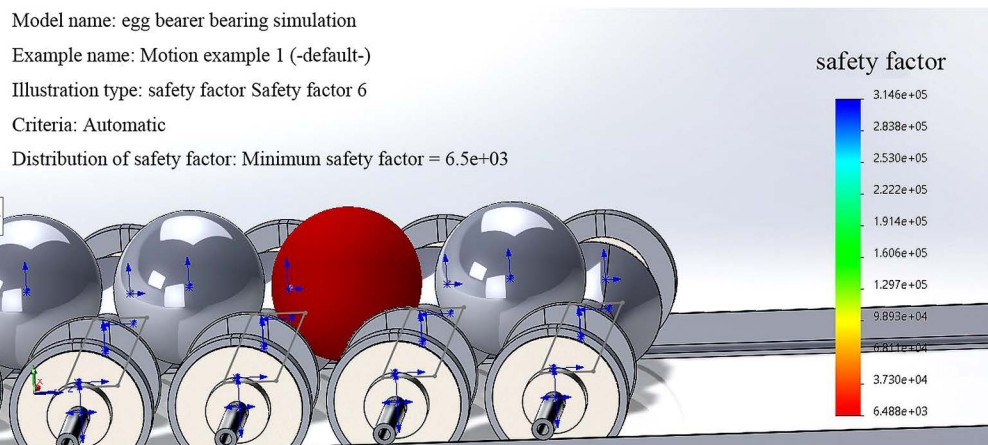

**Fig 18. Safety factors for eggs during transportation.**

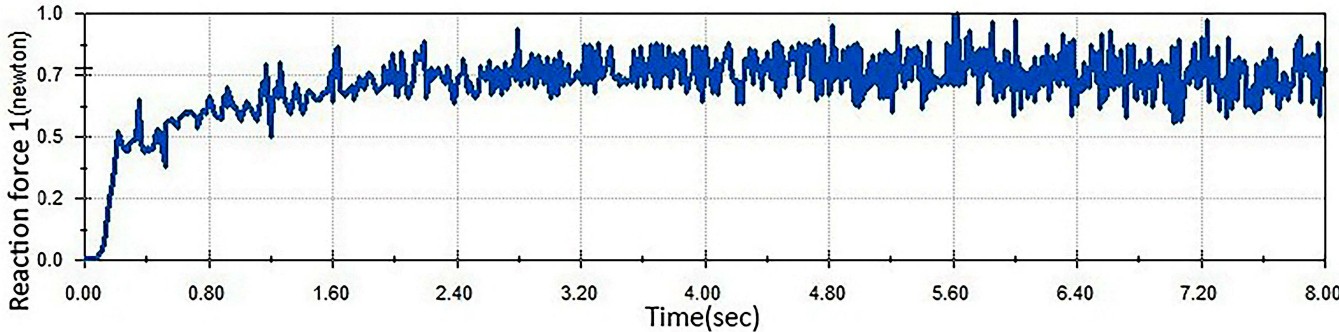

**Fig 19.   Contact force profile of eggs in the improved egg carriers.**

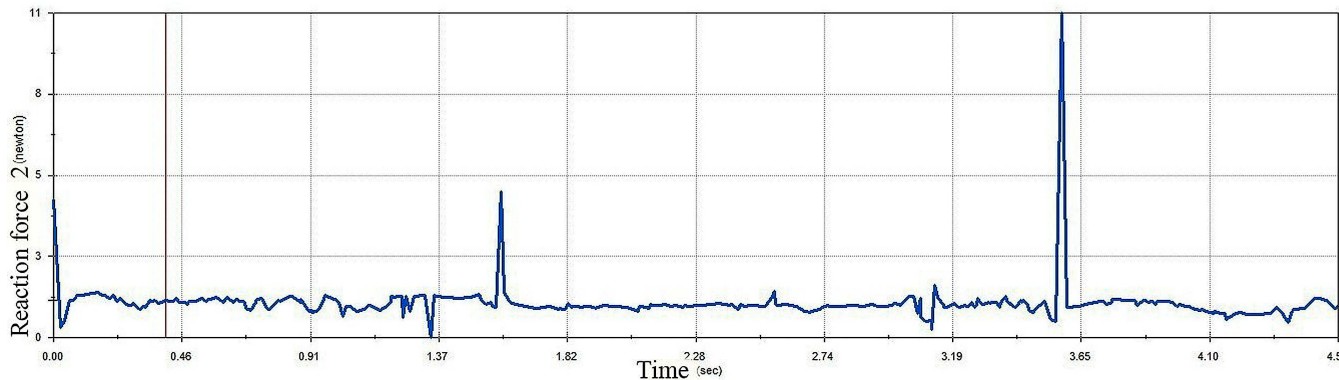

**Fig 20.   Contact force on eggs during transport in old egg carrier.**

In the comparison experiment, the old egg carrier performed relatively poorly. As shown in the Fig 20, the old egg carrier presented a sudden increase in contact force during the transport of eggs, indicating that the decrease in the number of broken eggs during transport was caused by this phenomenon. In contrast, the maximum contact force on the surface of the egg during transport of the improved egg carrier does not exceed 1 N, which is within the range of the tolerable contact force of the egg; therefore, the improved egg carrier can transport the eggs safely. Compared with the previous model, the new egg carrier mitigated this problem, effectively solving the problem of cracked egg damage.

The design of this improved egg carrier needs to address not only transport issues but also the natural rolling of the egg during crack detection or cleaning processes. The current method for distinguishing whether eggs have crack defects by analysing acoustic signals requires precise and controllable egg rolling. To verify whether the improved egg carrier can meet the design requirements, the relationship between the two egg carriers and the angular velocity of the egg was analysed, as shown in Figs 21 and 22.

The results show that the old egg carrier is not as good as it should be in terms of transport controllability. During transport, the angular velocity is unstable, so the rolling of the egg is not well controlled during the actual production process. In contrast, the angular velocity of the egg is essentially the same as that of the improved egg carrier, which verifies that the improved egg carrier can achieve more accurate control of the natural rolling of the egg.

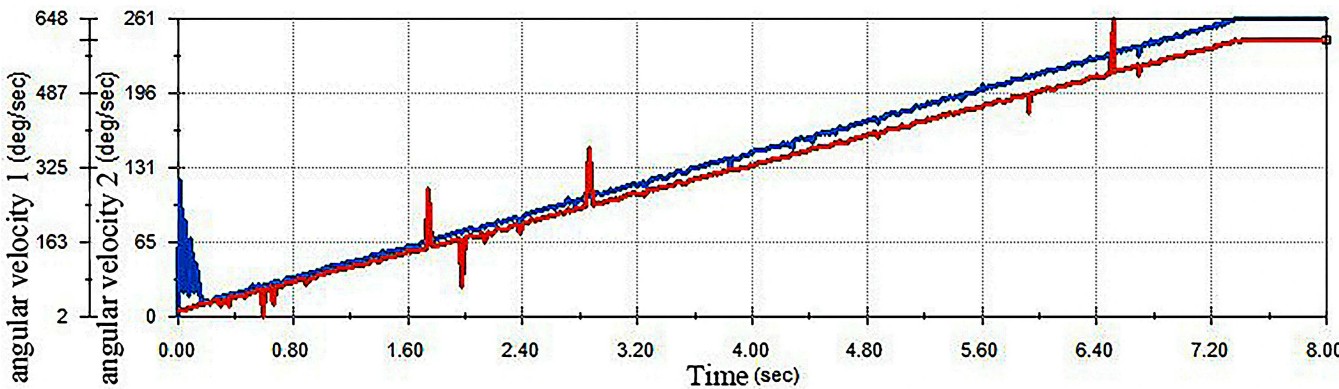

**Fig 21.  Angular velocity of eggs versus angular velocity of improved egg carriers.**

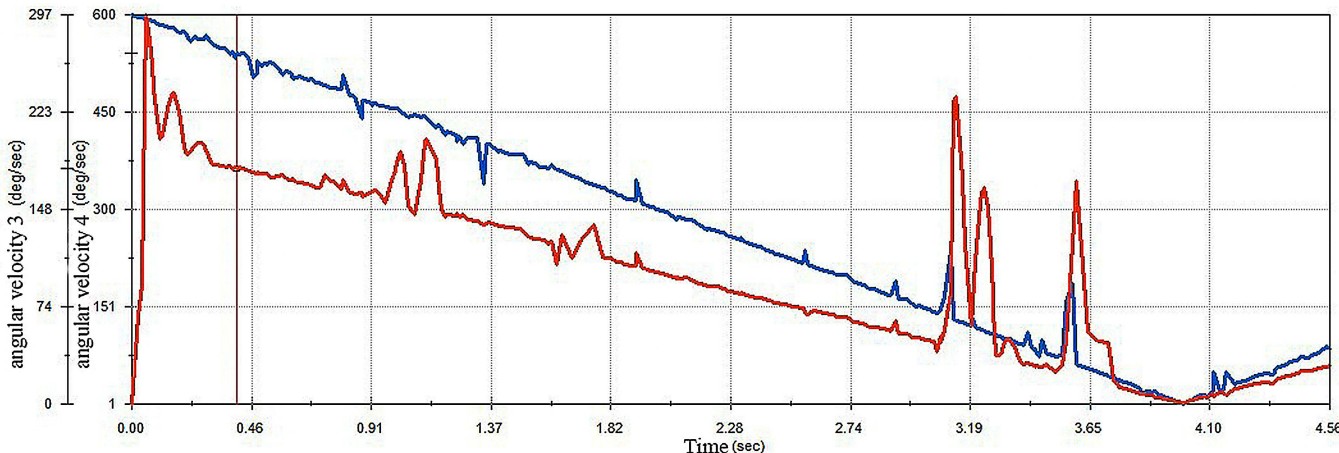

**Fig 22.  Angular velocity of eggs versus angular velocity of old egg carriers.**

To verify that the improved egg carrier is superior to the older egg carrier in terms of transport stability, an experiment was designed to compare the position of the centre of mass of the egg during transport. The two egg carrier transport processes were compared via kinematic analysis, keeping the angular velocity of the egg carrier at 580 deg/s and the position of the centre of mass of the egg in the y-axis direction as the study variable. The stability of the transport process was analysed, as shown in Figs 23 and 24.

As shown in the Figs 23 and 24, the old egg carrier shakes within approximately 1 mm in the y-axis direction during transport, whereas the improved egg carrier is significantly more stable. Therefore, the improved egg carrier is better than the old carrier in terms of transport stability and meets the design requirements.

Table 6 can be derived from several simulation comparison experiments between the improved egg carrier and the old egg carrier.

Table 6 shows that the improved egg carrier has been optimized for numerous parameters and is superior to the old egg carrier in terms of volume, mass, overall cost, transport rate and probability of defects. The optimized design meets the design requirements.

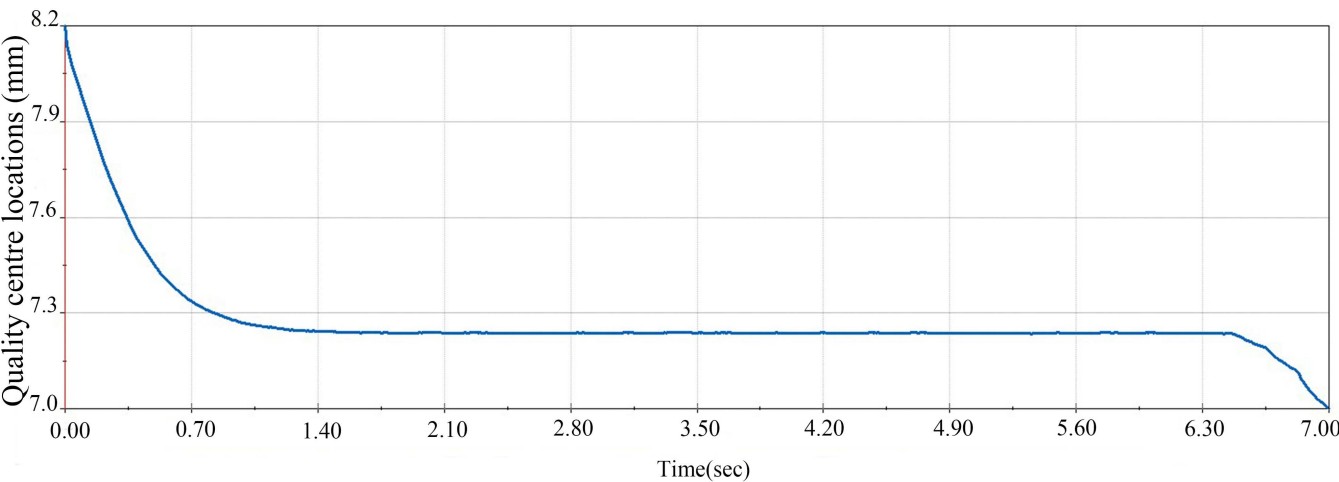

**Fig 23. Changes in the centre of mass of the egg during transport with the improved egg carrier.**

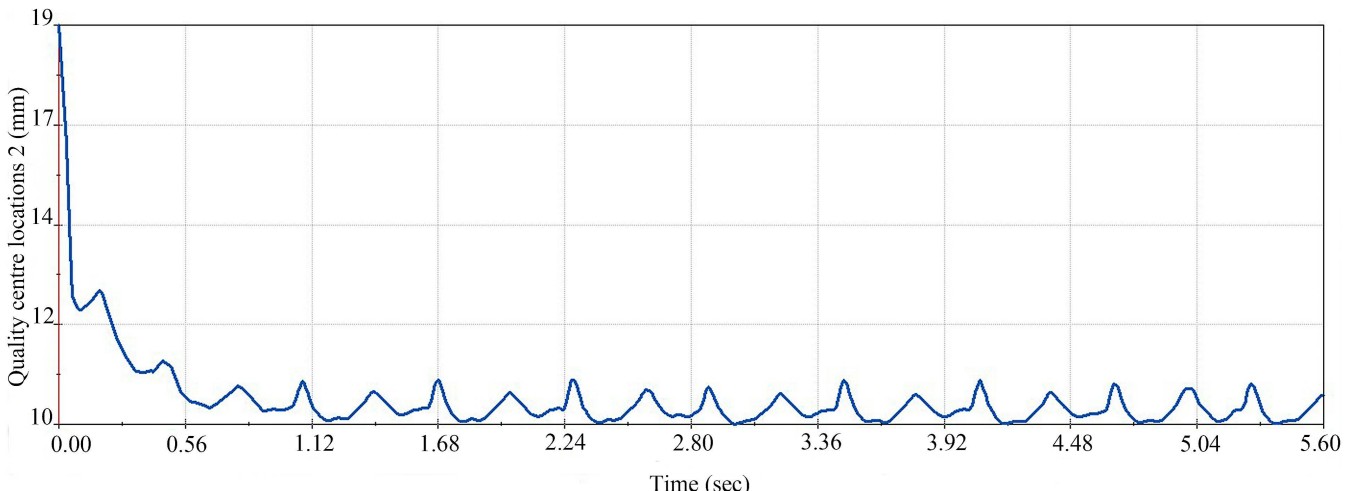

**Fig 24. Changes in the centre of mass of the egg during transport with the old egg carrier.**

## 7. Managerial insights

The structural optimization techniques presented in this study offer significant managerial implications for engineering design and project management. The following sections detail the applicability of the model in real-life engineering settings, the benefits to a company, and the ease of implementation in practice.

First, the optimized design of the egg carrier not only enhances the efficiency of the production process but also reduces material and manufacturing costs, leading to substantial economic benefits. Second, the use of the genetic algorithm as a decision-making tool provides a robust framework for evaluating different design alternatives, enabling engineers to make informed decisions that balance performance, cost, and reliability. Additionally, the finite element analysis (FEA) and fatigue analysis results offer valuable insights into the

**Table 6. Simulation experiment results.**

| Optimization parameters | Improved egg carrier | Old egg carrier |
|---|---|---|
| Egg tray volume | 0.0449 $m^3$ | 0.0531 $m^3$ |
| Egg tray quality | 49.44 g | 58.43 g |
| Egg tray cost (excluding moulds) | 2.68 $yuan$ | 3.99 $yuan$ |
| Overall cost (excluding moulds) | 21.88 $yuan$ | 33.99 $yuan$ |
| Maximum transport speed | 3,500 $deg/s$ | 1,200 $deg/s$ |
| Number of experiments | 40 | 40 |
| Total egg carrying capacity | 240 | 240 |
| Number of eggs dropped during the experiment | 1 | 18 |
| Number of broken eggs during the experiment | 0 | 8 |

structural integrity of the egg carrier, helping to identify potential failure points and mitigate risks during the design phase. For example, in the food processing industry, the efficient and stable transportation of eggs is crucial for maintaining product quality and reducing waste. The optimized egg carrier design ensures that eggs are transported with minimal contact force, reducing the risk of cracks and breaks. This proactive approach to risk management can significantly reduce the likelihood of costly rework and downtime in the production environment. Finally, the kinematic simulation results highlight the improved stability and transport efficiency of the optimized egg carrier, which can lead to better product quality and higher customer satisfaction.

The improved egg carrier also has many benefits for the company. First, it reduces costs. The optimized design reduces material and manufacturing costs. The use of food-grade epoxy resin for the egg tray and the plug-in structure of the egg carrier rod and friction wheel make the production process more efficient and cost-effective. Second, it improves the quality of the product. The reduced contact force and stable transport mechanism minimize the risk of egg defects, leading to increased product quality and customer satisfaction. Finally, it improves productivity. The improved egg carrier can operate at relatively high speeds without compromising stability, increasing the overall production rate and throughput.

Moreover, the improved egg carrier is easier to use in practice. The improved egg carrier also has many benefits for the company. First, it reduces costs. Second, it improves the quality of the product. Finally, it improves productivity.

Moreover, the improved egg carrier is easier to use in practice. First, the modular design makes it easy to assemble and maintain. The use of the genetic algorithm for optimisation and finite element analysis for validation ensures that the design is robust and reliable. These techniques are well established in the field of engineering and can be easily applied by experienced engineers.

Overall, the study findings provide a comprehensive guide for managers and engineers looking to implement advanced optimization techniques in their design and production processes.

# 8. Conclusions, limitations, and recommendations for future research

## 8.1. Findings

On the basis of the results and discussion, the following conclusions are drawn.

(1) The GA is used to solve the optimization design problem of the egg tray structure of the egg carrier, and a better optimization design effect is achieved. The results show that the improved egg carrier has obvious improvements in stability and efficiency. In addition, the production volume of the improved egg carrier is optimized by 18.18% compared with that of the existing egg carrier.

(2) Structural analysis of the improved egg carrier, finite element analysis, fatigue analysis and other validation analyses were performed. The strength, stiffness and service life of this component can meet the requirements of actual use.

(3) Kinematic simulation of the improved egg carrier was used to analyse the maximum speed at which the component works properly under normal working conditions. The stresses on the eggs during transport were also simulated to verify that there would be no stress concentration during transport.

(4) A simulation comparison experiment was designed to demonstrate the performance of the improved egg bearer compared with the older optimized model through data plots, and the results of this optimization study were visualized in detail via tables.

## 8.2.  Research Limitations

Despite the advancements presented in this study, several limitations should be acknowledged. First, the research focused primarily on the structural optimization of egg carriers for dynamic egg slit detection platforms. While the genetic algorithm and finite element analysis provided robust optimization and validation, the study did not extensively explore the long-term wear and tear of the materials used in the egg carrier. The mechanical properties of materials such as EN AW-2007 aluminium alloy, nylon 101, and food-grade epoxy resin were assumed to remain constant over the operational lifespan of the egg carrier. However, in real-world applications, these materials may experience degradation over time, which could affect the performance and reliability of the egg carrier.

Second, the study's experimental validation was conducted through computational simulations rather than physical prototypes. While the simulations provided valuable insights into the performance of the improved egg carrier, the absence of physical testing means that certain practical factors, such as environmental influences and manufacturing tolerances, were not considered. These factors could impact the real-world performance of egg carriers.

Finally, the focus on a specific type of egg carrier may limit the generalizability of the findings. The optimization techniques and design improvements presented may not be directly applicable to other types of egg carriers or similar transportation systems used in different industries.

## 8.3.  Recommendations for future research

Future research should address the limitations identified in this study and further explore the potential of the optimization techniques presented. First, long-term durability testing of the materials used in the egg carrier should be conducted to assess their performance over extended periods of use. This could involve accelerated ageing tests and cyclic loading experiments to simulate real-world operational conditions. The results of these tests could inform the selection of more durable materials or the implementation of protective coatings to increase the lifespan of the egg carrier.

Second, future work should include the development and testing of physical prototypes of improved egg carriers. This would allow for a more comprehensive evaluation of its performance, including the effects of environmental factors and manufacturing variations. Physical

testing could also reveal any unforeseen issues that may arise during operation, providing valuable data for further design refinements.

Third, a systems-level approach should be adopted to investigate the integration of the improved egg carrier with other components of the egg slit detection platform. This could involve simulations and experiments to understand how the egg carrier interacts with the platform's sensors, actuators, and control systems. The goal would be to optimize the overall performance of the platform and ensure seamless operation in various industrial settings.

Finally, future research should explore the applicability of optimization techniques and design improvements to other types of egg carriers and transportation systems. This could involve adapting the genetic algorithm and finite element analysis methods to different structural designs and operational requirements. The findings could then be generalized to benefit a wider range of applications in food processing and other industries.

By addressing these limitations and recommendations, future research can build upon the foundations laid by this study and further advance the field of structural optimization for dynamic egg slit detection platforms.

## Author contributions

**Conceptualization:** Ronghua Meng, Yuxiang Tian.

**Data curation:** Yuxiang Tian.

**Formal analysis:** Yuxiang Tian.

**Funding acquisition:** Ronghua Meng.

**Methodology:** Yuxiang Tian, Siwei Huang.

**Project administration:** Yuxiang Tian.

**Software:** Yuxiang Tian.

**Supervision:** Ronghua Meng, Siwei Huang.

**Validation:** Yuxiang Tian.

**Visualization:** Yuxiang Tian.

**Writing – original draft:** Yuxiang Tian.

**Writing – review & editing:** Ronghua Meng, Yuxiang Tian, Siwei Huang.

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
