## [Decision Letter · Decision Letter 0]

14 Jan 2025

PONE-D-24-54693Structural Design and Optimization of Egg Carrier for Dynamic Egg Slit Detection PlatformsPLOS ONE

Dear Dr. Tian,

Thank you for submitting your manuscript to PLOS ONE. After careful consideration, we feel that it has merit but does not fully meet PLOS ONE’s publication criteria as it currently stands. Therefore, we invite you to submit a revised version of the manuscript that addresses the points raised during the review process.

**ACADEMIC EDITOR:**

1. Please add some references in 2024 and 2025 if they exist.

2. Make the research gap section clear and highlight your work, separately.

3. Figure 3, Figure 11, Figure 12, Figure 13,19,20,21… shouldn’t be screenshots. Improve the quality.

4. Check the English presentation of this paper to remove the typo mistakes. Some grammatical issues need to be addressed in the whole text. Please reform the long paragraphs. Please polish the writing and English of the manuscript carefully. The writing of the paper needs a lot of improvement in terms of grammar, spelling, and presentation. The paper needs careful English polishing since there are many typos and poorly written sentences. I

5. The introduction should cover the following points:

· Background and Context: Provide a brief background of the problem. Discuss the current challenges and issues in this area.

· Research Gap and Significance: Highlight the gap in the existing literature and explain why it is important to address the problem. Emphasize the significance of the study in contributing to the existing body of knowledge.

· Objectives and Scope: Clearly state the research objectives and outline the study's scope. This should include a brief overview of the methodology and approach that will be used to achieve the objectives.

· Structure of the Paper: Provide a roadmap of the paper by outlining the sections and subsections that will be covered, giving the reader a clear understanding of what to expect in the subsequent parts of the paper.

6. Improve your " Conclusion" section. It should have been there. This part should be divided into 3 parts: Findings, Research limitations, and Recommendations for future research. (separately section)

7. Due to the high volume of calculations, all the formulas should be re-checked to ensure that there are no errors in terms of indices, typing, or concepts.

8. Check all of your Figures and Tables have a good explanation of your text.

9. Managerial implications are missing from the paper.

10. Check all the references that are correct and not duplicated.

11. Please emphasize the applicability of your model in a real-life engineering setting; give examples. What benefits would your paper bring to a company? How easy is it to implement it in practice? Please add a case study section to your paper if it does not exist.

12. You can use the suggested structure for your article:

1. Introduction

2. Literature review

2.1 Related studies

2.2 Research gap analysis and contributions

3. Problem description

4. Solution approach

5. Computational experiment and case study

6. Sensitivity analysis

7. Managerial insights

8. Conclusions, limitations, and recommendations for future research

8.1 Findings

8.2Research Limitations

8.3 Recommendations for future research

Create a document containing all of your appropriate clear answers. I am going with a major revision at this stage and waiting for your corrections. Then, I will give you my technical comments. Please use the yellow highlight after revising.

We look forward to receiving your revised manuscript.

Kind regards,

Sunny Narayan

Academic Editor

PLOS ONE

Journal Requirements:

“This work is financially supported by the National Natural Science Foundation of China (Grant No. 51975324, Grant No. 52075292) and the Natural Science Foundation of Hubei Province (Grant No. 2022CFC033).”

5. We note that your Data Availability Statement is currently as follows: All relevant data are within the manuscript and its Supporting Information files.

6. We note you have included a table to which you do not refer in the text of your manuscript. Please ensure that you refer to Table 6 in your text; if accepted, production will need this reference to link the reader to the Table.

7. We notice that your supplementary tables are included in the manuscript file. Please remove them and upload them with the file type 'Supporting Information'. Please ensure that each Supporting Information file has a legend listed in the manuscript after the references list.

Reviewers' comments:

Reviewer's Responses to Questions

**Comments to the Author**

1. Is the manuscript technically sound, and do the data support the conclusions?

Reviewer #1: Yes

2. Has the statistical analysis been performed appropriately and rigorously? 

Reviewer #1: I Don't Know

3. Have the authors made all data underlying the findings in their manuscript fully available?

Reviewer #1: Yes

4. Is the manuscript presented in an intelligible fashion and written in standard English?

Reviewer #1: Yes

5. Review Comments to the Author

Reviewer #1: 1. Please add some references in 2024 and 2025 if they exist.

2. Make the research gap section clear and highlight your work, separately.

3. Figure 3, Figure 11, Figure 12, Figure 13,19,20,21… shouldn’t be screenshots. Improve the quality.

4. Check the English presentation of this paper to remove the typo mistakes. Some grammatical issues need to be addressed in the whole text. Please reform the long paragraphs. Please polish the writing and English of the manuscript carefully. The writing of the paper needs a lot of improvement in terms of grammar, spelling, and presentation. The paper needs careful English polishing since there are many typos and poorly written sentences. I

5. The introduction should cover the following points:

· Background and Context: Provide a brief background of the problem. Discuss the current challenges and issues in this area.

· Research Gap and Significance: Highlight the gap in the existing literature and explain why it is important to address the problem. Emphasize the significance of the study in contributing to the existing body of knowledge.

· Objectives and Scope: Clearly state the research objectives and outline the study's scope. This should include a brief overview of the methodology and approach that will be used to achieve the objectives.

· Structure of the Paper: Provide a roadmap of the paper by outlining the sections and subsections that will be covered, giving the reader a clear understanding of what to expect in the subsequent parts of the paper.

6. Improve your " Conclusion" section. It should have been there. This part should be divided into 3 parts: Findings, Research limitations, and Recommendations for future research. (separately section)

7. Due to the high volume of calculations, all the formulas should be re-checked to ensure that there are no errors in terms of indices, typing, or concepts.

8. Check all of your Figures and Tables have a good explanation of your text.

9. Managerial implications are missing from the paper.

10. Check all the references that are correct and not duplicated.

11. Please emphasize the applicability of your model in a real-life engineering setting; give examples. What benefits would your paper bring to a company? How easy is it to implement it in practice? Please add a case study section to your paper if it does not exist.

12. You can use the suggested structure for your article:

1. Introduction

2. Literature review

2.1 Related studies

2.2 Research gap analysis and contributions

3. Problem description

4. Solution approach

5. Computational experiment and case study

6. Sensitivity analysis

7. Managerial insights

8. Conclusions, limitations, and recommendations for future research

8.1 Findings

8.2Research Limitations

8.3 Recommendations for future research

Create a document containing all of your appropriate clear answers. I am going with a major revision at this stage and waiting for your corrections. Then, I will give you my technical comments. Please use the yellow highlight after revising.

6. PLOS authors have the option to publish the peer review history of their article (what does this mean? ). If published, this will include your full peer review and any attached files.

**Do you want your identity to be public for this peer review?** For information about this choice, including consent withdrawal, please see our Privacy Policy .

Reviewer #1: No

---

## [Author Response · Author response to Decision Letter 1]

24 Feb 2025

ACADEMIC EDITOR:

1. Please add some references in 2024 and 2025 if they exist.

After a careful search of the relevant references, I found articles on optimising the structure of mechanical components by genetic algorithms, but I didn't find any specific references on optimising the egg tray by genetic algorithms. At the same time, I found Chinese references on the structural design of egg trays, but after careful reading, I found that the data in them are very problematic, so I don't think it is applicable.

2. Make the research gap section clear and highlight your work, separately.

The research gaps have been highlighted in the introductory section and the subsequent work I have done has been clearly articulated.

3. Figure 3, Figure 11, Figure 12, Figure 13,19,20,21… shouldn’t be screenshots. Improve the quality.

Images have been modified

4. Check the English presentation of this paper to remove the typo mistakes. Some grammatical issues need to be addressed in the whole text. Please reform the long paragraphs. Please polish the writing and English of the manuscript carefully. The writing of the paper needs a lot of improvement in terms of grammar, spelling, and presentation. The paper needs careful English polishing since there are many typos and poorly written sentences. I

I'm sorry for this problem, please understand that my first language is not English and I have touched it up at AJE.

5. The introduction should cover the following points:

· Background and Context: Provide a brief background of the problem. Discuss the current challenges and issues in this area.

· Research Gap and Significance: Highlight the gap in the existing literature and explain why it is important to address the problem. Emphasize the significance of the study in contributing to the existing body of knowledge.

· Objectives and Scope: Clearly state the research objectives and outline the study's scope. This should include a brief overview of the methodology and approach that will be used to achieve the objectives.

· Structure of the Paper: Provide a roadmap of the paper by outlining the sections and subsections that will be covered, giving the reader a clear understanding of what to expect in the subsequent parts of the paper.

The Introduction section has been revised.

6. Improve your " Conclusion" section. It should have been there. This part should be divided into 3 parts: Findings, Research limitations, and Recommendations for future research. (separately section)

The conclusion section has been reworked and the sections Research limitations and Recommendations for future research have been added.

7. Due to the high volume of calculations, all the formulas should be re-checked to ensure that there are no errors in terms of indices, typing, or concepts.

All formulas and calculation processes have been rechecked.

8. Check all of your Figures and Tables have a good explanation of your text.

Images and tables give a better explanation of my content

9. Managerial implications are missing from the paper.

Managerial implications have been added.

10. Check all the references that are correct and not duplicated.

I've checked all the references

11. Please emphasize the applicability of your model in a real-life engineering setting; give examples. What benefits would your paper bring to a company? How easy is it to implement it in practice? Please add a case study section to your paper if it does not exist.

This section has been added to the Managerial implications

12. You can use the suggested structure for your article:

1. Introduction

2. Literature review

2.1 Related studies

2.2 Research gap analysis and contributions

3. Problem description

4. Solution approach

5. Computational experiment and case study

6. Sensitivity analysis

7. Managerial insights

8. Conclusions, limitations, and recommendations for future research

8.1 Findings

8.2Research Limitations

8.3 Recommendations for future research

The structure of the article has been modified according to the proposed structure, but regarding the sensitivity analysis part, I think that the research I did was mainly about the optimisation of the structure of the egg-bearing apparatus, which has a simpler structure and does not involve the adjustment of the sensitive data, so I did not add this part in the process of modification.

---

## [Editor Report · Decision Letter 1]

26 Feb 2025

Structural Design and Optimization of Egg Carrier for Dynamic Egg Slit Detection Platforms

PONE-D-24-54693R1

Dear Dr. Tian,

We’re pleased to inform you that your manuscript has been judged scientifically suitable for publication and will be formally accepted for publication once it meets all outstanding technical requirements.

Kind regards,

sunny narayan

Academic Editor

PLOS ONE
---

## [Editor Report · Acceptance letter]

PONE-D-24-54693R1

PLOS ONE

Dear Dr. Tian,

I'm pleased to inform you that your manuscript has been deemed suitable for publication in PLOS ONE. Congratulations! Your manuscript is now being handed over to our production team.

Kind regards,

on behalf of

Dr. sunny narayan

Academic Editor

PLOS ONE